# Validation study of Boil & Spin Malachite Green Loop Mediated Isothermal Amplification (B&S MG-LAMP) versus microscopy for malaria detection in the Peruvian Amazon

Keare A. Barazorda[1☯], Carola J. Salas[2☯], Greys Braga[2], Leonila Ricopa[2], Julia S. Ampuero[3], Crystyan Siles[3], Juan F. Sanchez[2], Silvia Montano[2], Stephen E. Lizewski[2], Christie A. Joya[2], Danett K. Bishop[2], Hugo O. Valdivia[2]*

1 Vysnova Partners Inc, Lima, Peru, 2 Department of Parasitology, U.S. Naval Medical Research Unit N˚6 (NAMRU-6), Lima, Peru, 3 Department of Virology and Emerging infections, U.S. Naval Medical Research Unit N˚6 (NAMRU-6), Lima, Peru

☯ These authors contributed equally to this work.
* hugo.o.valdivia.ln@mail.mil

## Abstract

Malaria elimination efforts in Peru have dramatically reduced the incidence of cases in the Amazon Basin. To achieve the elimination, the detection of asymptomatic and submicroscopic carriers becomes a priority. Therefore, efforts should focus on tests sensitive enough to detect low-density parasitemia, deployable to resource-limited areas and affordable for large screening purposes. In this study, we assessed the performance of the Malachite–Green LAMP (MG-LAMP) using heat-treated DNA extraction (Boil & Spin; B&S MG-LAMP) on 283 whole blood samples collected from 9 different sites in Loreto, Peru and compared its performance to expert and field microscopy. A real-time PCR assay was used to quantify the parasite density. In addition, we explored a modified version of the B&S MG-LAMP for detection of submicroscopic infection in 500 samples and compared the turnaround time and cost of the MG-LAMP with microscopy. Compared to expert microscopy, the genus B&S MG-LAMP had a sensitivity of 99.4% (95%CI: 96.9%– 100%) and specificity of 97.1% (95%CI: 91.9%– 99.4%). The *P. vivax* specific B&S MG-LAMP had a sensitivity of 99.4% (96.6%– 100%) and specificity of 99.2% (95.5%– 100%) and the *P. falciparum* assay had a sensitivity of 100% (95%CI: 78.2%– 100%) and specificity of 99.3% (95%CI: 97.3%– 99.8%). The modified genus B&S MG-LAMP assay detected eight submicroscopic malaria cases (1.6%) which the species-specific assays did not identify. The turnaround time of B&S MG-LAMP was faster than expert microscopy with as many as 60 samples being processed per day by field technicians with limited training and utilizing a simple heat-block. The modified B&S MG-LAMP offers a simple and sensitive molecular test of choice for the detection of submicroscopic infections that can be used for mass screening in resources limited facilities in endemic settings nearing elimination and where a deployable test is required.

**Data Availability Statement:** All relevant data are within the manuscript and its Supporting Information files.

**Funding:** This work was supported by the US DoD Armed Forces Health Surveillance Division (AFHSD), Global Emerging Infections Surveillance (GEIS) Section, PROMIS IDs P0106_18_N6_02 (DKB) and P0143_19_N6_02 (CAJ) from 2018 and 2019. The funders had not role in the study design, data collection and analysis, preparation of the manuscript nor decision to publish. KAB is an employee of Vysnova and this funder provided support in the form of salaries for KAB, but did not have any additional role in the study design, data collection and analysis, decision to publish, or preparation of the manuscript. The specific roles of this author is articulated in the author contributions' section.

**Competing interests:** KAB is an employee of Vysnova and reports personal fees outside the submitted work. This does not alter our adherence to PLOS ONE policies on sharing data and materials.

# Introduction

Malaria is a major problem worldwide with more than 228 million cases and 405,000 deaths in 2018 [1]. Malaria prevention and control efforts have reduced malaria incidence in several countries. However, achieving the elimination goal would require the development of improved diagnostic methods and tools capable of detecting low-density infections that can complement existing interventions [2].

In Peru, it is estimated that at least one third of the population live in malaria endemic areas and are at risk of contracting the disease [3]. Malaria cases are concentrated in the Department of Loreto in the Peruvian Amazon which accounts for 96% of all cases reported in the country during 2017 [4]. Malaria cases are predominantly in rural and riverine communities with very little communication and access to health care.

In this context, the Peruvian Ministry of Health (MoH) developed the plan "Malaria Zero" which is aimed towards malaria elimination in the Amazon region [3,5]. These efforts have dramatically reduced the incidence of cases in the Amazon basin making the detection of asymptomatic and submicroscopic carriers critical in the final stages of elimination as they are known to sustain transmission [6]. Therefore, efforts should focus on tests that are capable of detecting low-density parasitemia, can be performed in resource-limited areas, and are inexpensive enough to screen large populations [2].

Malaria diagnosis in Peru is performed through routine microscopy and rapid diagnostic tests (RDTs) [7]. However, both methods present some challenges to malaria elimination goals because of their inability to detect low parasite density infections. Microscopy requires experienced personnel to provide quality results and is not reliable for detecting low parasite density infections since its limit of detection is about 50 parasites/μL [8–10]. The drawbacks of RDTs are also lower sensitivity with limit of detection of 100–200 parasites/μL [11]. In addition, in Peru there is a relatively high prevalence of parasites that have deleted the *P. falciparum*-specific genes HRP2 and HRP3 prevalent in up to 41% and 70% of *P. falciparum* samples, respectively [12] limiting their ability to detect falciparum infections. These limitations make the development of highly sensitive, cost-effective malaria diagnostic tests highly valuable for Peru and similar malaria elimination programs in South America. Early malaria detection will also improve patient care and patient outcomes [1].

Loop-mediated isothermal amplification (LAMP)-based assays were developed to meet the need for a sensitive field-usable assay for numerous diseases including malaria [13–19]. Furthermore, the addition of dyes such as malachite-green facilitates colorimetric readout, thus reducing cost and simplifying the assays.

The Malachite-Green Loop-mediated Isothermal Amplification (MG-LAMP) has shown great performance in field and laboratory-controlled settings [20–22]. We recently demonstrated that this method can be implemented in a reference laboratory in Peru and we documented its usefulness for surveillance studies in areas approaching malaria elimination where submicroscopic infections are challenging to detect by conventional methods like microscopy or RDTs [23].

The optimization of molecular assays for use in resource-limited areas requires simple DNA extraction methods as opposed to conventional DNA extraction kits that are more expensive and require specialized equipment and laboratory space. Methods such as extraction through heat–treated whole blood are a promising alternative method [24,25].

In this study, we assessed the performance and cost of MG-LAMP using heat-treated whole blood (Boil and Spin or B&S) DNA extraction for detection of *P. falciparum* and *P. vivax* malaria in field conditions and compared its performance to microscopy as gold standard. A real-time PCR assay (PET-PCR) was used for parasite quantification.

## Methods

### Study sites and sample collection

The study was carried out in nine study sites in Loreto region which is located in the northwest side of the Peruvian Amazon basin (Fig 1). Samples for this project were obtained from 3 retrospective studies: two passive surveillance studies of febrile disease etiology conducted between 2015 and 2019 in Iquitos, Peru and surroundings communities and from an active surveillance study conducted in 2019 in the Iquitos community of Padre Cocha.

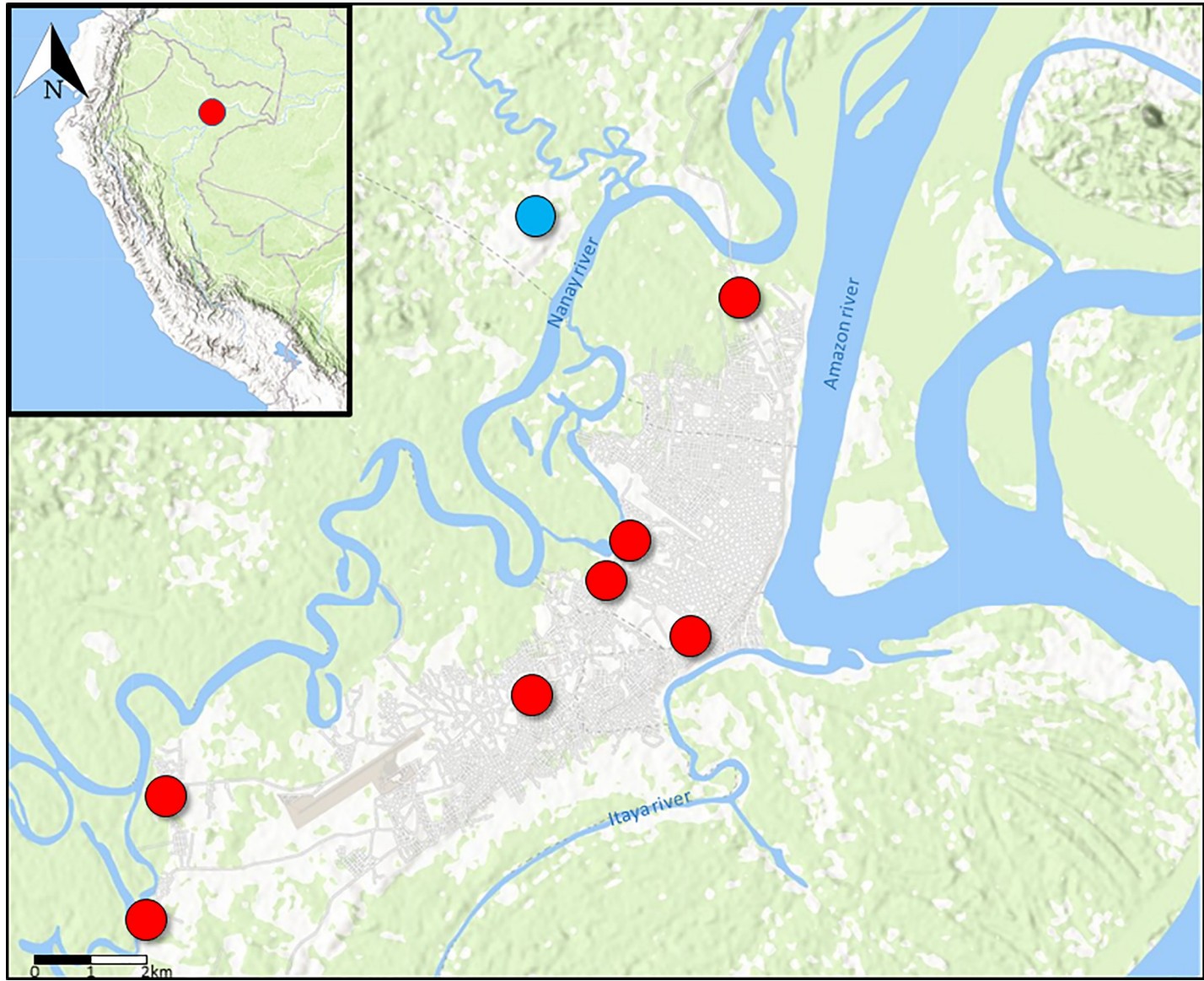

**Fig 1. Study sites located in the region of Loreto, Peru.** The insert shows the location of the health facilities where samples were collected in Iquitos. Red circles showed the sites for the passive surveillance studies. The community of Padre Cocha (Blue) was a site for the passive and active surveillance study. Map created using open data from OpenStreetMap, under ODbL, under CC BY 2.0 (creativecommons.org/licenses/by/2.0/).

## Sample size

The sample size was calculated using the formula for a single proportion assuming a sensitivity of 96% for the MG-LAMP based on previous studies and 95% confidence and 80% power. This resulted in a minimum sample size of 196 samples to be tested.

## Passive surveillance studies

A total of 283 whole blood samples were selected from two studies, one conducted in 2015–2019 and another in 2018. The first study was designed for malaria surveillance and 258 samples were used. Inclusion criteria for this study were age greater than 1 year with presence of fever or history of fever during the previous 72 hours and signed informed consent to participate in the study (and assent where appropriate).

The second passive surveillance study was designed for febrile diseases etiology and 25 samples were used. Inclusion criteria included patients 5 years of age or older who presented in outpatient clinics or hospitals with acute, undifferentiated, febrile illness (axillary temperature greater than or equal to 38°C for 7 days duration or less) with specific symptoms [26].

## Active surveillance study

Samples from the active surveillance study were collected between July 15th to August 23th 2019 from a cohort study aimed at detecting asymptomatic/submicroscopic malaria infections in the community of Padre Cocha (Fig 1). Inclusion criteria for this study were age greater than or equal to 5 years and resident of Padre Cocha or surroundings areas. Only expert microscopy was available to process all samples from this study.

## Ethical considerations

The samples tested in this study were selected from three protocols (NMRCD.2007.0004, NMRCD.2010.0010 and NAMRU6.2018.0008) previously approved by the Institutional Review Board of the U.S Naval Medical Research Unit N˚6 (NAMRU-6) in compliance with all applicable federal regulations governing the protection of human subjects. Protocols were also approved by the local Peruvian MoH. Adult participants provided written consent and minors provided written assent and parental/guardian consent.

## Training of laboratory staff to perform MG-LAMP in the field site

A standard operating procedure (SOP) was prepared in the reference laboratory at NAMRU-6, Lima and provided to the technicians working in our Iquitos facility before onsite training.

Training on MG-LAMP was provided on-site to two NAMRU-6-Iquitos technicians with basic experience on molecular methods. Before testing the B&S MG-LAMP in the field site, training was conducted in two sessions for two days. The first session was theoretical to address the LAMP chemistry, differences between nested-PCR and advantages of LAMP approaches. The second session was practical, with a first guided experiment by the instructor using positive and negative malaria controls. The second experiment was executed by the technicians under the instructor's supervision using samples collected during the week.

## Sample processing and microscopy

Two levels of microscopy were performed in this study, field and expert microscopy. Two mL of EDTA-whole blood was collected by venipuncture from each participant. Expert microscopy procedure consisted of two thin and thick smears, stained with 10% Giemsa and read by two fully dedicated microscopists in the laboratory at NAMRU-6 Iquitos. A slide was

considered negative after reading 200 oil-immersion fields while positive slides were quantified using as reference 6000 WBC/μl of blood. The remaining whole blood was aliquoted in one or two vials and stored at -80˚C for further testing. Field microscopy was performed at health-care facilities of each community as per the health facilities' guidelines. Thick blood smears were prepared, slides were stained for 10 minutes with 10% Giemsa, and 100 fields per slide were read. Results were recorded using the "plus system" following the Peruvian MoH guidelines [7,27].

## DNA extraction methods

Two DNA extraction methods were performed for all the samples tested, i) boil and spin method (B&S) was used only for the MG-LAMP and ii) Qiagen™ kit was used for PET-PCR and nested-PCR. The B&S DNA extraction method was performed as previously described [24,25,28] with a minor modification in the elution step to concentrate the obtained DNA. Briefly, an aliquot of 60 μL of whole blood was transferred into a 1.5 mL tube containing 60 μL of extraction buffer (400 mM NaCl, 40 mM Tris pH 6.5, 0.4% SDS). The mixture was incubated for 5 min using a heat-block set at 95˚C followed by centrifugation for 3 min at 10,000g. Finally, we prepared a 1:4 dilution consisting of 30 μL of clear supernatant mixed with 90 μL of ultra-pure water. Extracted DNA was immediately used or stored at -20˚C for no longer than a week. DNeasy Blood & Tissue kit (Qiagen)™ was used for DNA extraction for PET-PCR method. All samples were processed at NAMRU-6 Lima following the manufacturer's instruction using 200 μL of EDTA-whole blood. DNA was eluted in 70μL of elution buffer and used immediately for the amplification reactions; the remaining volume was stored at -20˚C.

## Malachite green LAMP assay (MG-LAMP)

For the passive surveillance studies MG-LAMP was performed using the B&S DNA (B&S MG-LAMP). The primers used are listed in S1 Table. All samples were first tested with the genus-specific MG-LAMP assay. The genus positive samples were further tested by the species-specific assays for both *P. falciparum* and *P. vivax*. Each assay was carried out using individually capped tube strips to avoid cross-contamination among samples. The final volume of each reaction was 20 μL, which contained 2X in-house LAMP buffer (40 mM Tris-HCL pH 8.8, 20 mM KCL, 16 mM MgSO$_4$, 20 mM (NH$_4$)SO$_4$, 0.2% Tween-20, 1.5M Betaine, 2 mM of DNTP's each, 0.004% Malachite Green dye (MG), 320U/mL of Bst DNA polymerase (New England Biolabs, Ipswich, MA) and 5 μL of B&S DNA template, as previously described [20].

Samples collected from the active surveillance study were processed using a modified version of the original MG-LAMP master mix in a final volume of 25 μL containing 2X in-house LAMP buffer, 0.0048% Malachite Green dye (MG), 320U/mL of Bst DNA polymerase and 10 μL of B&S DNA template (1.6x of target DNA in comparison with the MG-LAMP used on the passive surveillance component).

To maintain the Quality Control (QC) of the in-house buffer, a set of positive controls were tested every time a new LAMP buffer batch was prepared before testing field samples. All amplification reactions were done for 1 hour at 63˚C using a mini heat block (Gene Mate, Bio Express, Utah, US). PCR reaction tube strips were removed and kept at room temperature for 15 minutes before being scored for amplification by two independent readers. A positive result was defined by the presence of a light blue-green color, while the negative samples were colorless. If the color obtained after amplification was grey or uncertain, a second reaction was performed.

### Photo-induced electron transfer-PCR (PET-PCR)

To quantify parasite density and assess the limit of detection of the B&S MG-LAMP, all samples were tested in duplicates using a *Plasmodium* genus PET-PCR as previously described with a slight modification on the cut-off value [29]. All genus positive samples were further tested by singleplex reactions to detect *P. falciparum* and *P. vivax* as previously described [21,29]. Samples were considered positive if they had a threshold cycle (Ct) of 41 or below and negative if they had no Ct value or a Ct value above 41. This cut-off was determined using clinical samples.

### Nested-PCR

The 18srRNA-nested PCR was chosen for species identification of specimens collected in the active surveillance study because of its high sensitivity and specificity [30].The first reaction amplified the 18srRNA subunit ribosomal (ssrRNA) gene and the second reaction amplified specific regions for each plasmodium species (*P. falciparum and P. vivax*). Both PCR reactions were performed in 50 μl of final volume. For the first reaction 5 μl of DNA template was used, while 5 μl of the amplified product were used in the second reaction. Results were visualized in a 2% agarose gel stained with GelRed®

### Analytical sensitivity of B&S MG-LAMP

To assess the limit of detection (LoD) of the B&S MG-LAMP we used serial dilutions *Plasmodium* control strains. For *P. falciparum*, we performed continuous *in vitro* culture of the 3D7 reference strain following the same procedure previously described [23]. For *P. vivax* we used a clinical specimen with 121,500 parasites/μL. For both controls, we performed tenfold dilutions starting from 10,000 parasites/μL to 1 parasite/μL with two-fold dilutions from 10 parasites/μL to 1.25 parasites/μL. These controls were also used for training laboratory staff in Iquitos.

### Turn-around time and cost analysis

We used 8 samples to estimate the required processing time, turn-around time (TAT) and costs of expert microscopy and B&S MG-LAMP for the specific identification of *Plasmodium* genus and species. Cost-analysis included the costs required for sample preparation (preparation of slides and staining for microscopy) and DNA extraction and amplification supplies (for B&S MG-LAMP). Salaries were not included in the calculations.

### Quality assessment

To evaluate the limitations and accuracy of our study, we followed the guidelines of the Quality Assessment of Diagnostic Accuracy Studies (QUADAS-2) which evaluate four areas of bias (patient selection, index test, gold standard method, and flow/timing) [31]. Concerns regarding applicability were also evaluated in three areas: patient selection, index test, and reference standards and where also evaluated to have low, unclear or high risk of bias.

### Statistical methods

Sensitivity and specificity with 95% confidence intervals (95% CI) were calculated for field microscopy and B&S MG-LAMP using expert microscopy results as the gold standard test. Statistical differences between expert microscopy and B&S MG-LAMP were determined using the non-parametric McNemar test. The Cohen's Kappa coefficient [32] was used to estimate agreement between diagnostic tests and the area under the receiver operating characteristic

(ROC) curve was used to compare the performance between tests. Analysis of PET-PCR quantification was performed on log-transformed data to compare with microscopy quantification. Data were analyzed using Stata, version, 13 (StataCorp LP, College station, TX).

## Results

### Analytical limit of detection of B&S MG-LAMP

The analytical LoD for the genus B&S MG-LAMP was one parasite/μL while that of the *P. falciparum* and *P. vivax* was 2.5 parasites/μL for both assays with 100% concordance between two independent readers (Fig 2). Regarding the modified B&S MG-LAMP used in the active surveillance component, the LoD for the genus and species-specific assays was 0.5 parasites/μL.

### Malaria species detection in the passive surveillance study

A total of 283 samples were used in this study (S2 Table): All the 283 samples were tested with the *Plasmodium* genus- specific MG-LAMP using DNA obtained by the B&S extraction method. A total of 67 samples were tested at NAMRU-6 laboratory in Lima, while the remaining 216 were tested at field laboratory in Iquitos. A flow chart of all laboratory tests performed is shown in S1 Fig).

As summarized in Table 1, field microscopy identified 109 negatives and 174 positives (18 *P. falciparum* and 156 *P. vivax*). Expert microscopy identified 105 negatives and 178 positives (15 *P. falciparum*, 162 *P. vivax* and 1 mixed *P. falciparum/P. vivax*) with a mean asexual

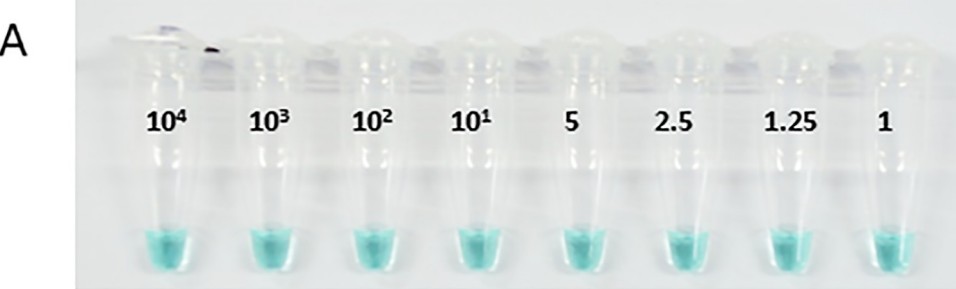

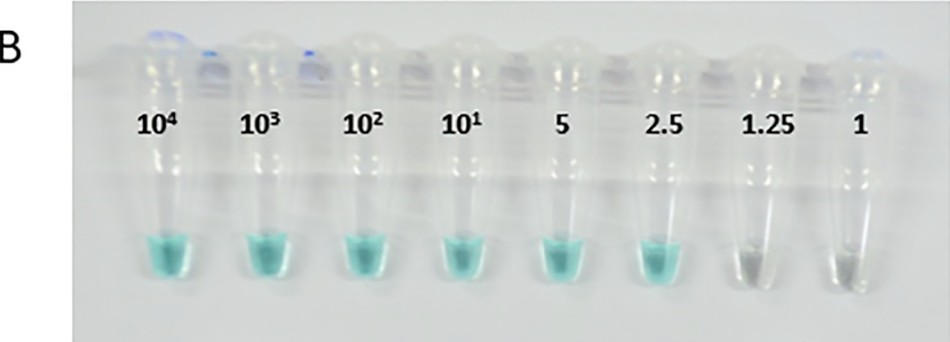

**Fig 2. Analytical limit of detection of B&S MG-LAMP assays.** The analytical limit of detection of genus (A) and *P. vivax* (B) B&S MG-LAMP was determined using 10-fold ($10^4$–1.0 parasite/μL) and 2-fold (5–1.25 parasite/μL) serial dilution of well-quantified *P. falciparum* (genus) and *P. vivax* samples, as indicated on the tubes. Colorimetric B&S MG-LAMP results can be visualized with the naked-eye, blue/green indicate positive whereas colorless, negative. NC = no-template control. The LoD of the genus-specific B&S MG-LAMP assay was 1 parasite/μL and that of *P. vivax* was 2.5 parasite/μL.

**Table 1. Summary of *Plasmodium* detection results obtained for each method.**

|  | Field Microscopy | Expert Microscopy | B&S MG- LAMP | nested-PCR | PET-PCR |
|---|---|---|---|---|---|
| *P. vivax* | 156 | 162 | 162 | 170 | 168 |
| *P. falciparum* | 18 | 15 | 17 | 11 | 11 |
| Pf/Pv infections | 0 | 1 | 0 | 6 | 9 |
| Indeterminate species | 0 | 0 | 1 | 0 | 4 |
| Negative | 109 | 105 | 103 | 96 | 91 |
| **Total** | **283** | **283** | **283** | **283** | **283** |

Pf: *P. falciparum*, Pv: *P. vivax*.

parasitemia of 796 parasites/μL (range: 12–69,235 parasites/μL) and mean gametocytemia of 72.46 gametocytes/μL (range: 12–1,235 gametocytes/μL).

The genus B&S MG-LAMP identified 180 *Plasmodium spp* and 103 negatives whereas the species-specific B&S MG-LAMP identified 179 positives (17 *P. falciparum* and 162 *P. vivax*).

## Diagnostic performance of microscopy and B&S MG-LAMP

Using expert microscopy as gold standard, field microscopy had a sensitivity for detecting either *Plasmodium spp.* of 97.2% (95%CI: 93.6–99.1%) and specificity of 99% (95%CI: 94.8% - 100%). Regarding the species-specific detection, field microscopy had a sensitivity for *P. vivax* of 96.3% (95%CI: 92.1%– 98.6%) and specificity of 100% (95%CI: 97%– 100%). For *P. falciparum*, field microscopy had a sensitivity of 93.3% (95%CI: 68.1%– 99.8%) and specificity of 98.5% (95%CI: 96.2%– 99.6%) (Fig 3, S3 Table). The genus B&S MG-LAMP had a sensitivity of 99.4% (95%CI: 96.9%– 100%) and specificity of 97.1% (95%CI: 91.9%– 99.4%). The *P. vivax* specific B&S MG-LAMP had a sensitivity of 99.4% (96.6%– 100%) and specificity of 99.2% (95.5%– 100%) and the *P. falciparum* assay had a sensitivity of 100% (95%CI: 78.2%– 100%) and specificity of 99.3% (95%CI: 97.3%– 99.8%) (Fig 3, S3 Table).

## Agreement of B&S MG-LAMP and field microscopy compared with expert microscopy

The McNemar test did not show statistical differences between the results of the B&S MG-LAMP and expert microscopy for the detection of *Plasmodium spp*, *P. vivax* and *P.*

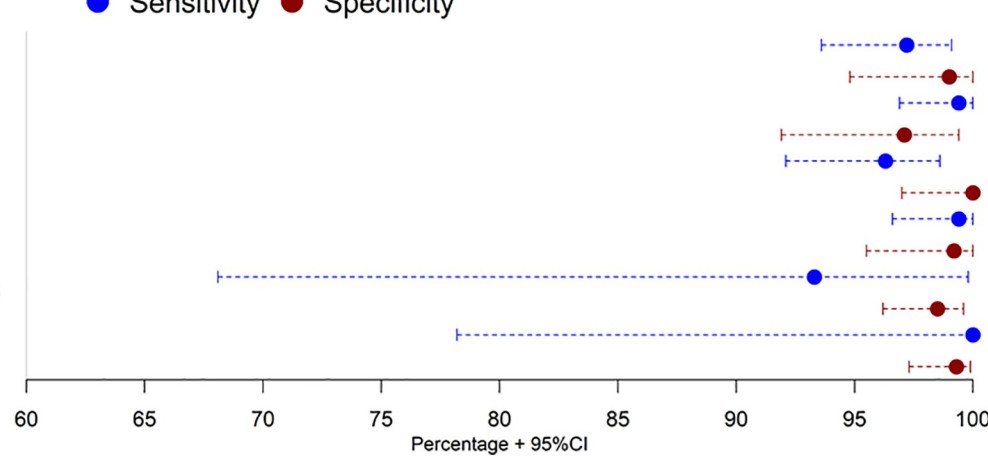

**Fig 3. Sensitivity (blue), specificity (red) and 95% confidence intervals for genus and species detection of field microscopy, expert microscopy and the B&S LAMP.**

**Table 2. Results of positive from the active surveillance study.** The remaining 493 samples were negative.

| Samples | Expert microscopy | Genus B&S MG-LAMP | Nested PCR/PET-PCR | Genus PET-PCR parasites/ uL |
|---|---|---|---|---|
| 1 | *P. vivax* | Positive | *P. vivax* | 14.7 |
| 2 | Negative | Positive | *P. vivax* | 9.4 |
| 3 | Negative | Positive | *P. vivax* | 3.3 |
| 4 | Negative | Positive | *P. vivax* | 45.0 |
| 5 | Negative | Positive | *P. falciparum* | 5.2 |
| 6 | Negative | Positive | *P. vivax* | 1.3 |
| 7 | Negative | Positive | *P. vivax* | 6.3 |
| 8 | Negative | Positive | *P. vivax* | N/A |

N/A: Not available = sample was not available for testing by PET-PCR.

*falciparum* (p> 0.05). However, statistical differences were found between field microscopy and B&S MG-LAMP for the detection of *P. vivax* (p<0.05), but there were no differences for the detection of genus and *P. falciparum*. Cohen's kappa showed that B&S MG-LAMP presented a kappa of 0.97 (95%CI: 0.94–0.99) and field microscopy of 0.93 (95%CI: 0.89–0.98). The area under the ROC curve for the B&S MG-LAMP was 0.99 while for field microscopy was 0.97 (S2 Fig).

## Discordant results

Of the 178 positive samples, nine were shown to be mixed infections by PET-PCR. Expert microscopy detected one mixed infection which was shown to be *P. falciparum* infection by the other methods. The B&S MG-LAMP and field microscopy detected eight and seven, respectively, of the nine mixed infections as single infections, detecting only the predominant species (S4 Table). The median species parasitemia in the mixed infections as determined by PET-PCR was 60.3 (0.2–72 parasites/uL) for *P. falciparum* and 0.7 (0.3–10,830 parasites/ uL) for *P. vivax*. Eight out of the 9 missed mixed infections had parasite densities below the limits of detection of the B&S MG-LAMP and microscopy (S4 Table). In addition, eight samples diagnosed as *P. vivax* by the PET-PCR were found to be negative by the B&S MG-LAMP and field microscopy while expert microscopy detected only one of these (S4 Table). The mean parasitemia for these samples was 0.6 parasites/uL.

## B&S MG-LAMP performance in the active surveillance study

A total of 500 participants were enrolled on the active surveillance study with a mean age of 33 ± 20 years old and 57% were females. Malaria-like symptoms in the previous three days before enrollment were reported by 21% of participants. These symptoms included headache (85%), malaise (39%), fever (29%) and chills (28%). Expert microscopy detected one *P. vivax* case (0.2%) while the genus B&S MG-LAMP assay detected eight positive cases (1.6%) (Table 2). The range of parasites/mL of the 8 positive samples was 1.3–45.0 parasites/μL as determined by genus PET-PCR. Nested-PCR identified these as 7 *P. vivax* and 1 *P. falciparum* (Table 2). However, the species-specific B&S MG-LAMP did not identify any positive samples. One out of eight positive subjects reported fever during the previous 3 days before enrollment (12.5%).

## Turnaround time (TAT) and Cost Analysis

For ease of estimation, the TAT and cost analysis was estimated using 8 samples because the strip of PCR tubes used in the MG-LAMP has 8 samples (Table 3). The processing of a total of

**Table 3. Cost and TAT of B&S MG-LAMP assay and expert microscopy for 8 samples (One strip of tubes).**

| | Microscopy | B&S MG-LAMP |
|---|---|---|
| Cost (USD)[1] | $7.52 | $ 9.60 |
| TAT (hours)[2] | 4 | 2 |
| Hands-on work (hours)[3] | 3 | 1 |

[1]Estimated cost includes reagents and materials for DNA extraction and slide preparation without considering labor costs.

[2]TAT is the time required from samples processing to getting results

[3]Hands-on work is the actual time that a technician spent at the laboratory performing the procedure.

8 samples was about $2 cheaper by expert microscopy compared to B&S MG-LAMP and the overall TAT was shorter by B&S MG-LAMP compared to expert microscopy (Table 3). The total number of samples the technicians were able to process per day was evaluated during the processing of samples from the active surveillance study: the technicians increased the number of samples processed daily by B&S MG-LAMP from 12 samples to 60 samples per day. All 500 samples were processed in 12 working days by two lab technicians.

## Discussion

Malaria transmission in several countries in the Americas has been gradually declining due to renewed efforts towards malaria elimination [1]. Under this context, there is a great need for highly sensitive tools to detect asymptomatic or low parasite density carriers that are frequently missed by current RDTs and microscopy [12,33–35]. Furthermore, these new tools need to be tailored to the epidemiology and conditions of endemic settings in each country so that they can better complement and inform targeting of malaria interventions [2].

In this study, we assessed the performance of a simplified MG-LAMP using heat-treated whole blood versus field and expert microscopy. Our results from the passive surveillance studies showed that overall, the B&S MG-LAMP has a similar performance compared to both levels of microscopy. This can be explained by the fact that the mean asexual parasitemia in the passive surveillance study was 796 parasites/μL which is within the detection limits of both microscopy and MG-LAMP. In addition, the malaria control program in Peru has invested heavily in quality microscopy even in health centers/diagnostic post greatly improving the quality of microscopy even in the field settings [3]. This suggests that both levels of microscopy are good options for case management in Peru. In contrast, the modified B&S MG-LAMP used in the active surveillance study was superior to field and expert microscopy in the detection of asymptomatic low-density infections.

The use of PET-PCR for parasite quantification allowed us to estimate the parasitemia on the positive genus samples missed by the species B&S MG-LAMP on the passive surveillance component. Our results showed that in all these cases, parasite densities were below the analytical LoD of the B&S MG-LAMP. Furthermore, in the case of missed mixed infections, the B&S MG-LAMP detected the parasite with a higher parasite density and missed the other species as they occurred at extremely low-parasite densities, below the detection limit of B&S MG-LAMP. However, it is also possible that this failure was caused by a limitation of the primer target used in the MG-LAMP or the method of signal detection as previously observed [19,21,23].

These are important issues to address since misdiagnosis in mixed infections can lead to inappropriate treatment management in addition to increased relapse risk for *P. vivax* infections as previously shown [36]. Our results showed that false negative samples by B&S

MG-LAMP were classified as submicroscopic infections by PET-PCR. This scenario highlights the need for continuing developing methods for highly sensitive detection which are key in areas undergoing elimination such as Peru. In these regions, accurate detection of submicroscopic cases prevents ongoing transmission by tackling hotspots of submicroscopic and asymptomatic reservoirs that are responsible of continuing transmission [3,37]. Nonetheless, the modified B&S MG-LAMP assay tested on the active surveillance study has the potential for detection of submicroscopic and asymptomatic infections. However, more thorough testing of its performance is needed.

Heat-treatment has been utilized in previous studies as an alternative to the more expensive DNA extraction kits for LAMP [38–40]. The LoD of the genus and species-specific B&S MG-LAMP assays observed in this study was similar to that observed using the QIAGEN DNA extraction kit in previous studies [13,23]. This shows that the use of the B&S unpurified DNA extraction does not impact the performance of the MG-LAMP reaction dye [41]. In addition, unlike Taq polymerase, *Bst* polymerase used in LAMP is not affected by inhibitors of conventional PCR such as haemoglobin [42,43], making LAMP easily adaptable to easier DNA extraction methods such as B&S. However, the B&S DNA template should be utilized soon after the extraction in order to avoid enzyme degradation [38]. Our B&S DNA extraction protocol included a final DNA concentration step during the elution step which improved the performance of the MG-LAMP compared to the original protocol [21,44].

The TAT of B&S MG-LAMP was faster than expert microscopy with as many as 60 samples being processed per day by field technicians with limited training and utilizing a simple heat-block. With additional heat-blocks, this number of samples can be increased to allow the screening of many more samples. The cost per sample of the B&S MG-LAMP was calculated to be about $1.2 compared to $0.94 for expert microscopy. This cost is much lower compared to other molecular methods such as PCR-based assays [45]. Therefore, the B&S MG-LAMP offers a test of choice for mass screening and detection of submicroscopic infections due to i) minimal expertise required, ii) reduced cost iii) potential for increased throughput with portable heat-blocks for large scale screening which is a limitation for microscopic diagnosis and iv) increased sensitivity over field and expert microscopy as shown in our active surveillance study.

Using the QUADAS-2 tool, we evaluated that there was a low bias risk for sample identification by the operators as they did not have access to individual sample identification nor study file details. For the reference B&S MG-LAMP test, samples were tested in batches of 30 specimens at the end of the week for the passive surveillance studies or 1 month after enrolment of the active surveillance.

In summary, our study shows that the modified B&S MG-LAMP is a simpler molecular test that can be implemented in resources limited facilities in endemic settings nearing for elimination and where a deployable test is required [19] such as points of entry and in support of active surveillance studies. The increasing availability of such tests in the field will help in the detection of parasite reservoirs that can be targeted with interventions to help interrupt transmission. Our study also highlights the need of maintaining good malaria microscopy skills that can deliver quality results. We validated a methodology that aimed to be compliant with the WHO [46], to solve the urgent need for new tools to support the eradication efforts.

## Supporting information

**S1 Fig. Flow chart of all methods performed in the study.**
(TIF)

**S2 Fig. ROC curve comparison with expert microscopy.**
(TIF)

**S1 Table. List of primers used for the B&S MG-LAMP.**
(DOCX)

**S2 Table. Individual test results for passive surveillance component.**
(XLSX)

**S3 Table. Comparison of the B&S MG-LAMP and field microscopy with expert microscopy.**
(DOCX)

**S4 Table. Mixed infections results.** Comparison of results of mixed infections (Pf/Pv) performed by each methodology.
(DOCX)

## Acknowledgments

We would like to thank Dr. Venkatachalam Udhayakumar for his support in reviewing the article and express our gratefulness to Naomi W. Lucchi for her support in conceptualization, writing and reviewing the draft and added valuable suggestions to obtain this final version. She and Dragan Ljolje also worked in improving the current methodology. Finally, we thank the laboratory team in Iquitos for their continuous hard work in collecting samples and to the anonymous volunteers who kindly contributed to this outcome.

## Ethical approval

The study protocols were approved by the Naval Medical Research Center Unit 6 Institutional Review Board in compliance with all applicable Federal regulations governing the protection of human subjects.

**Disclaimer:** The views expressed in this article reflect the results of research conducted by the author and do not necessarily reflect the official policy or position of the Department of the Navy, Department of Defense, nor the United States Government.

## Author Contributions

**Conceptualization:** Stephen E. Lizewski, Hugo O. Valdivia.

**Data curation:** Greys Braga.

**Formal analysis:** Keare A. Barazorda, Carola J. Salas, Hugo O. Valdivia.

**Funding acquisition:** Danett K. Bishop.

**Methodology:** Keare A. Barazorda, Carola J. Salas, Greys Braga, Leonila Ricopa.

**Supervision:** Stephen E. Lizewski.

**Writing – original draft:** Keare A. Barazorda, Carola J. Salas, Julia S. Ampuero, Crystyan Siles, Juan F. Sanchez, Silvia Montano, Stephen E. Lizewski, Christie A. Joya, Danett K. Bishop, Hugo O. Valdivia.

**Writing – review & editing:** Keare A. Barazorda, Carola J. Salas, Greys Braga, Leonila Ricopa, Julia S. Ampuero, Crystyan Siles, Juan F. Sanchez, Silvia Montano, Stephen E. Lizewski, Christie A. Joya, Danett K. Bishop, Hugo O. Valdivia.

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
