## [Decision Letter · Decision Letter 0]

2 Jul 2021

PONE-D-21-14913

Validation study of Boil & Spin Malachite Green Loop Mediated Isothermal Amplification (B&S MG-LAMP) versus microscopy for malaria detection in the Peruvian Amazon

PLOS ONE

Dear Dr. Valdivia,

Thank you for submitting your manuscript to PLoS ONE. After careful consideration, we felt that your manuscript requires revision, following which it can possibly be reconsidered. Although your manuscript was of interest to the reviewer, major concerns were related to study design and data presentation. According to the reviewers, the methods were not described in enough details to allow suitably skilled investigators to fully replicate and evaluate the study. Regarding the study design, the reviewer suggested that authors follow the recommendations of the STARD 2015 E QUADAS-2 tool. In addition, a significant number of issues should be clarified and/or adjust otherwise the MS’s results may be compromised. Finally, the authors should follow the policy of Plos One to share the raw data underlying their results.  Such policies help increase the reproducibility of the published literature, as well as make a larger body of data available for reuse and re-analysis. For your guidance, a copy of the reviewers' comments was included below 

We look forward to receiving your revised manuscript.

Kind regards,

Luzia Helena Carvalho, Ph.D.

Academic Editor

PLOS ONE

Journal Requirements:

"This work was supported by the US DoD Armed Forces Health Surveillance Division

(AFHSD), Global Emerging Infections Surveillance (GEIS) Branch, PROMIS IDs

P0106_18_N6_02 (DKB) and P0143_19_N6_02 (CAJ) from 2018 and 2019. The

funders had not role in the study design, data collection and analysis, prepation of the

manuscript nor decision to publish."

We note that one or more of the authors are employed by a commercial company: "Vysnova Partners Inc,"

Reviewers' comments:

Reviewer's Responses to Questions

**Comments to the Author**

1. Is the manuscript technically sound, and do the data support the conclusions?

Reviewer #1: Partly

Reviewer #2: Yes

2. Has the statistical analysis been performed appropriately and rigorously? 

Reviewer #1: Yes

Reviewer #2: Yes

3. Have the authors made all data underlying the findings in their manuscript fully available?

Reviewer #1: No

Reviewer #2: Yes

4. Is the manuscript presented in an intelligible fashion and written in standard English?

Reviewer #1: Yes

Reviewer #2: Yes

5. Review Comments to the Author

Reviewer #1: The manuscript about B&S MG-LAMP assay for the diagnosis of Malaria in Peruvian Amazon sounds interesting. The results obtained indicate that this method can be implemented in resources limited facilities in endemic settings nearing for elimination. However I identified several concerns that the AAs should evaluate carefully. I suggest that they should add more details to the methodology, mainly about of the design of study, and submit it for further evaluation.

Major comments

In the "Methods" section, authors need to improve writing regarding the study design. As this is an accuracy study, I suggest that the authors follow the recommendations of the STARD 2015 E QUADAS-2 tool. Including a flowchart with the study design may help. In addition, it is recommended that authors add a sample calculation. Considering the information present in line 144 ("The samples tested in this study were selected from three protocols"), this study was carried out using panel samples from 3 different studies. It is necessary to know the number of samples for each of the studies, as well as the definition of their status (cases or non-cases based on the reference test result). They must also inform the average time between collection and extraction of DNA from these samples. Apparently, the inclusion of patients was not blinded or consecutive and microscopy was used as index test and as standard reference. This can also introduce bias and may have consequences in data interpretation. I suggest a paragraph disclosing the main limitations of the present article according to the STARD 2015 E QUADAS-2 tool. Another critical point is that with the data presented by the authors, it is not possible to calculate all the values presented in the topic "Diagnostic performance of microscopy and B&S MG-LAMP". The results must be clarified by showing in the contingency table used for calculation of the statistical parameters. I also suggest including a supplementary table with all the results for each of the samples. Another suggestion is to perform the accuracy analysis including the PET-PCR test as a reference standard.

Minor comments

Line 247- include reference regarding the Cohen's Kappa coefficient.

Lines 278-79 - In the text, the authors reported that B&S MG-LAMP detected 180 positive samples, but in Table 1 there are 179 (17 P. falciparum and 162 P. vivax).

Lines 320-21 - "Eight samples diagnosed as P. vivax by the PET-PCR were found to be negative by the B&S MG-LAMP". Include in the "Discussion", the impact that false-negative LAMP results may have on malaria control programs.

Table 3- indicate that the cost calculation was estimated for 8 samples.

Reviewer #2: This manuscript describes interesting results that validate the simple boil and spin method for MG-LAMP based diagnosis of Plasmodium infections. Authors have covered almost all of required experiments to demonstrate performance of a new test including determination of assay turnaround time and cost evaluation. The authors claim that increasing the amount of target DNA in the reaction tube by a factor of 1.6 (ie 10 uL of sample per 25 uL reaction instead of 5 uL sample per 20 uL reaction as used in the simplified protocol) improves assay performance in cases of low density parasitaemia. However, no data is presented comparing the simplified and modified protocols within the same study population, say during the active surveillance study, to allow for assessment of cost-effectiveness of the modified protocol.

Minor comments:

1) Title: replace "malaria detection" with "malaria diagnosis"

2) L38-39: Replace "detect low-parasite densities" with "detect low-density parasitaemia"

3) in L232 it is stated that tenfold dilutions were done starting from 100 000 parasites/uL whereas in L262 and Fig 2 the starting parasite density is 10 000 p/uL

4) L256: replace ".....LoD...was one parasite/uL" with "...LoD was one parasite/uL or lower". The exact LoD is not known given that 1p/uL corresponds to the lowest density tested.

5) L322: Replace "seen" with "see"

6) Cost per reaction is stated as $1.2 in L408 and $9.60 in Table 3

6. PLOS authors have the option to publish the peer review history of their article (what does this mean?). If published, this will include your full peer review and any attached files.

Reviewer #1: **Yes: **Daniel Moreira de Avelar

Reviewer #2: No

---

## [Author Response · Author response to Decision Letter 0]

17 Aug 2021

RESPONSES

Comments to the Author

• Have the authors made all data underlying the findings in their manuscript fully available?

Reviewer #1: No

Response: We do not have any restrictions to provide an unidentified database. In this regard, we are including the individual tests results from the passive surveillance component on S2 table. Regarding the active surveillance database, individual results for the positive samples are described in Table 2. The remaining 493 samples of active surveillance were negative and we do not consider necessary to detail them in another supplementary table.

REVIEWER #1: 

Major comments

1. In the "Methods" section, authors need to improve writing regarding the study design. As this is an accuracy study, I suggest that the authors follow the recommendations of the STARD 2015 E QUADAS-2 tool. Including a flowchart with the study design may help. In addition, it is recommended that authors add a sample calculation. Considering the information present in line 144 ("The samples tested in this study were selected from three protocols"), this study was carried out using panel samples from 3 different studies. It is necessary to know the number of samples for each of the studies, as well as the definition of their status (cases or non-cases based on the reference test result). 

Response: We appreciate the reviewer’s suggestions. In this regard, we are providing a flowchart describing the process for the passive surveillance component on S1 fig. In addition, we included a paragraph regarding sample size calculation on the methods section. Regarding the studies and their status, the data is provided on Table 2 and S2 table. In summary, we have 258 (174 cases and 84 non-cases) from the first passive study, 25 subjects (4 cases and 21 non-cases) from the second passive study and 500 subjects (1 case and 499 non-cases) from the active surveillance. 

2. They must also inform the average time between collection and extraction of DNA from these samples. Apparently, the inclusion of patients was not blinded or consecutive and microscopy was used as index test and as standard reference. This can also introduce bias and may have consequences in data interpretation. I suggest a paragraph disclosing the main limitations of the present article according to the STARD 2015 E QUADAS-2 tool. 

Response: The average time between sample collection and DNA extraction for the passive surveillance studies were 5 months. On the other hand, the average time between collection and DNA extraction for the active surveillance study was 2 months. 

We used the STARD 2015 E and QUADAS-2 tools and found that there was a low bias in three areas. 

For the passive surveillance study, i) samples were randomly selected, ii) the staff that performed subject enrolment was different from the one performing microscopy, iii) B&S MG-LAMP was performed in batches of 30 at the end of the week and iv) molecular technicians were blinded to the individual study files.

For the active surveillance study, i) enrolment staff were different from the laboratory staff, ii) the technician doing expert microscopy was different from the one running the B&S MG-LAMP in most cases and iii) B&S MG-LAMP was performed at least one month after microscopy, iv) samples were processed in batches of 30-60 and the technician did not have access to the subject study files making very hard to remember microscopy results.

A text referring the use of QUADAS is currently available in the methods section and on the discussion.

3. Another critical point is that with the data presented by the authors, it is not possible to calculate all the values presented in the topic "Diagnostic performance of microscopy and B&S MG-LAMP". The results must be clarified by showing in the contingency table used for calculation of the statistical parameters. I also suggest including a supplementary table with all the results for each of the samples. Another suggestion is to perform the accuracy analysis including the PET-PCR test as a reference standard.

Response: Thank you for the suggestion. We are including S2 table with the individual results from the passive surveillance studies. For the active surveillance study, the results are presented in Table 2. We are not showing results from the remaining 493 samples since all of them were negative. Finally, table 1 now shows the results for all the laboratory tests. In this regard, calculations can be replicated using the information provided in any of both tables. 

4. Another suggestion is to perform the accuracy analysis including the PET-PCR test as a reference

Response: We appreciate the suggestion. However, the aim of the current work was to present the scientific community with a B&S MG-LAMP field deployable test and therefore it should be compared with other field deployable tests.

It is important to notice that we are planning to test the modified B&S MG-LAMP employed in the active study using PET PCR and nested PCR as reference. However, this work is still ongoing and samples are still pending to be collected.

Minor comments

5. Line 247- include reference regarding the Cohen's Kappa coefficient.

Response: Reference was included following your suggestion.

6. Lines 278-79 - In the text, the authors reported that B&S MG-LAMP detected 180 positive samples, but in Table 1 there are 179 (17 P. falciparum and 162 P. vivax).

Response: Thank you for the suggestion; we have updated table 1 and the text on this section for accuracy.

7. “Line 292 :The species-specific B&S MG-LAMP identified 179 positives (17 P. falciparum, 162 P. vivax) , the genus B&S MG-LAMP identified 1 Plasmodium spp and 103 negatives." 

Response: Thank you for the suggestion; we have corrected the information for clarity.

8. Lines 320-21 - "Eight samples diagnosed as P. vivax by the PET-PCR were found to be negative by the B&S MG-LAMP". Include in the "Discussion", the impact that false-negative LAMP results may have on malaria control programs.

Response: Thank you for the suggestion; we have included a paragraph in the discussion section.

9. Table 3-indicate that the cost calculation was estimated for 8 samples.

Response: Thank you for the suggestion, the title in Table 3 was updated indicating in the equivalence “8 samples = 1 strip of tubes”.

REVIEWER #2: 

Major comments

1. The authors claim that increasing the amount of target DNA in the reaction tube by a factor of 1.6 (ie 10 uL of sample per 25 uL reaction instead of 5 uL sample per 20 uL reaction as used in the simplified protocol) improves assay performance in cases of low density parasitaemia. However, no data is presented comparing the simplified and modified protocols within the same study population, say during the active surveillance study, to allow for assessment of cost-effectiveness of the modified protocol.

Response: Thank you for the insight provided for this manuscript. Regarding the modified B&S LAMP, we run a standard curve to document the limit of detection (LOD) and the estimated LOD was 0.5 parasites/µL for the genus and species-specific assays. Therefore, this result supports that this slight modification in master mix volumes and target DNA could potentially improve the assay performance for low parasitemia detection compared to the simplified protocol. However, we are aware that a more rigorous evaluation of this modified mixed is required and we are stating that in the discussion section of manuscript.

In this regard, the comparison of the original MG-LAMP master mix versus the new master mix with increased target DNA that you kindly suggested goes beyond the scope of this manuscript. Furthermore, a bigger sample size of low parasitemia/submicroscopic cases is needed for an accurate comparison both methods. We are aiming to obtain sufficient samples from upcoming follow-ups of the active cohort study.

Minor comments

2. Title: replace "malaria detection"with "malaria diagnosis"

Response: This change was performed as suggested. 

3. L38-39: Replace "detect low-parasite densities" with "detect low-density parasitaemia"

Response: Thank you for the suggestion, this change was updated as suggested.

4. in L232 it is stated that tenfold dilutions were done starting from 100 000 parasites/uL whereas in L262 and Fig 2 the starting parasite density is 10 000 p/uL

Response: Thank you for finding the error, we have corrected it to 10,000 p/uL.

5. L256: replace ".....LoD...was one parasite/uL" with "...LoD was one parasite/uL or lower". The exact LoD is not known given that 1p/uL corresponds to the lowest density tested.

Response: This change was corrected as suggested. 

6. L322: Replace "seen" with "see"

Response: Corrected as suggested.

7. Cost per reaction is stated as $1.2 in L408 and $9.60 in Table 3

Response: This is because 9.6 showed in table 3 is the cost per a strip of 8 samples. We have added a text in the legend for clarification.

---

## [Decision Letter · Decision Letter 1]

5 Oct 2021

Validation study of Boil & Spin Malachite Green Loop Mediated Isothermal Amplification (B&S MG-LAMP) versus microscopy for malaria detection in the Peruvian Amazon

PONE-D-21-14913R1

Dear Dr. Valdivia,

We’re pleased to inform you that your manuscript has been judged scientifically suitable for publication and will be formally accepted for publication once it meets all outstanding technical requirements.

Kind regards,

Luzia Helena Carvalho, Ph.D.

Academic Editor

PLOS ONE

Additional Editor Comments (optional):

Reviewers' comments:

Reviewer's Responses to Questions

**Comments to the Author**

1. If the authors have adequately addressed your comments raised in a previous round of review and you feel that this manuscript is now acceptable for publication, you may indicate that here to bypass the “Comments to the Author” section, enter your conflict of interest statement in the “Confidential to Editor” section, and submit your "Accept" recommendation.

Reviewer #1: All comments have been addressed

2. Is the manuscript technically sound, and do the data support the conclusions?

Reviewer #1: Yes

3. Has the statistical analysis been performed appropriately and rigorously? 

Reviewer #1: Yes

4. Have the authors made all data underlying the findings in their manuscript fully available?

Reviewer #1: Yes

5. Is the manuscript presented in an intelligible fashion and written in standard English?

Reviewer #1: Yes

6. Review Comments to the Author

Reviewer #1: The authors adequately addressed my comments raised in a previous round of review. Recommendation: Accept.

7. PLOS authors have the option to publish the peer review history of their article (what does this mean?). If published, this will include your full peer review and any attached files.

Reviewer #1: No

---

## [Editor Report · Acceptance letter]

14 Oct 2021

PONE-D-21-14913R1 

Validation study of Boil & Spin Malachite Green Loop Mediated Isothermal Amplification (B&S MG-LAMP) versus microscopy for malaria detection in the Peruvian Amazon 

Dear Dr. Valdivia:

I'm pleased to inform you that your manuscript has been deemed suitable for publication in PLOS ONE. Congratulations! Your manuscript is now with our production department. 

Kind regards, 

on behalf of

Dr. Luzia Helena Carvalho 

Academic Editor

PLOS ONE